# Medfluencer: A Network Representation of Medical Influencers' Identities and Discourse on Social Media

Zhijin Guo
zhijin.guo@bristol.ac.uk
University of Bristol
Bristol, UK

Edwin Simpson
edwin.simpson@bristol.ac.uk
University of Bristol
Bristol, UK

Roberta Bernardi
roberta.bernardi@bristol.ac.uk
University of Bristol
Bristol, UK

## Abstract

In our study, we first constructed a dataset from the tweets of the top 100 medical influencers with the highest Influencer Score [14] during the COVID-19 pandemic. This dataset was then used to construct a socio-semantic network, mapping both their identities and key topics, which are crucial for understanding their impact on public health discourse. To achieve this, we developed a few-shot multi-label classifier to identify influencers and their network actors' identities, employed BERTopic for extracting thematic content, and integrated these components into a network model to analyze their impact on health discourse. To ensure the reproducibility of our results, we have made the code available at https://github.com/ZhijinGuo/Medinfluencer.

## Keywords

social network, covid-19, social media, large language models

**ACM Reference Format:**
Zhijin Guo, Edwin Simpson, and Roberta Bernardi. 2024. Medfluencer: A Network Representation of Medical Influencers' Identities and Discourse on Social Media. In *Proceedings of the 7th epiDAMIK ACM SIGKDD International Workshop on Epidemiology meets Data Mining and Knowledge Discovery, August 26, 2024, Barcelona, Spain.* 8 pages.

## 1 Introduction

Medical influencers are individuals from either a medical or non-medical background who engage in health conversations on social media, and by doing so influence what the public think about a health issue. Social media posts by medical influencers can drive user engagement and a positive attitude change towards medical interventions [8] but may also present some risks by spreading misinformation online[7]. This has been shown by recent examples of online misinformation about the measles and COVID-19 vaccines and the consequent growth in vaccine hesitancy [17]. Therefore, understanding how medical influencers affect people's views about a health issue is of paramount importance to tackle misinformation and the growing mistrust in health authorities [18]. Our aim is to advance existing socio-semantic network analysis methods to investigate how medical influencers engage with other actors in their social network to influence discourse about a health issue on social media.

Socio-semantic network analysis connects actors' social relations with the relations between actors and the semantics of their words [3]. Existing socio-semantic network analysis methods have considered the centrality of actors' words in semantic networks and the centrality of actors' positions in social networks as the main source of influence of actors' discourse [13]. However, they neglect actors' identities as another important source of influence. Identities are sets of meanings that individuals attach to themselves to define 'who they are' in a role (e.g. 'doctor'), in a social group (e.g. 'female'), or as a person (e.g. 'being sociable') [5]. Individuals act and communicate in ways that are consistent with the meanings defining their identities. By defining who an individual is, identities also tell others what to expect from that individual and how to behave towards them [6] and are therefore a source of meaning that contributes to the formation of socio-cultural ties [9]. Indeed, research has shown how influencers' identities affect how communities of users engage with them and respond to their messages on social media [23]. Therefore, we advance existing socio-semantic network analysis methods to capture important indicators of influence of health discourse on social media, such as medical influencers' identities, i.e. how medical influencers categorise themselves in relation to a social group, and discursive frames, i.e. the meanings that medical influencers assign to a health issue. While identities define what medical influencers in a social network say and the resonance of their ideas with the actors they engage with [9], discursive frames represent how medical influencers seek to persuade the public about their view on a health issue [12].

In this paper, we present the following results of a project on Covid-19 Top 100 medical influencers on X (formerly Twitter):

1. A few-shot multi-label classifier of the identities of medical influencers and of actors in the influencers' social network, using few-shot methods to overcome the small number of labels per class.

2. A BERTopic model that identifies medical influencers' discursive frames from the influencers' tweets.

3. A map of relationships between medical influencers' identities, social network ties, and discursive frames.

Mapping socio-semantic network ties with actors' identities provides insights into how medical influencers adapt discourse to influence different audiences. These insights can prove useful for population and public health researchers and practitioners in the design of social media campaigns.

## 2 Data Collection and Labelling

*Data Collection.* We used the report from Onalytica [14] in order to identify the most influential healthcare professionals on X (formely Twitter). Onalytica uses a PageRank-based methodology to pinpoint influencers by evaluating the quality and quantity of

**Table 1: Users and Their Multi-Labels**

| User | Labels |
|------|--------|
| User1 | Scientist |
| User2 | Politician |
| User3 | Politician, Military member, Business professional |
| User4 | Media practitioner |
| User5 | Scientist, Physician |
| User6 | Scholar |
| User7 | Media practitioner |
| User8 | Cultural professional |
| User9 | Business professional, Media practitioner |
| User10 | Scholar, Educational professional, Cultural professional |

contextual references they receive, thus assessing their Topical Authority. This analysis includes examining social engagement on platforms like Twitter, and how frequently influencers are mentioned in COVID-19 related discussions on Instagram, Facebook, YouTube, forums, blogs, news, and Tumblr, considering their resonance, relevance, and reach.

We collected all tweets made from the official accounts of these healthcare professionals with package "academictwitteR" [2]. Due to the extensive number of tweets, we segmented the collection process into multiple queries, all within the same timeframe: from December 1, 2020, to February 24, 2022.

Subsequently, we compiled these tweets into a "tidy" format suitable for analysis in a data frame. This data frame includes various columns such as tweet ID and username, encompassing a total of 424,629 tweets.

*Data Labelling.* We randomly selected a sample of 2,000 users' bios from the 100 medical influencers and their followers. Two independent coders manually classified the users' self-categorizations or identities based on their bio sentences, and then met to resolve any disagreements. Of the 54 labels identified through the manual labelling, 34 were job occupations, 16 were organisation categories, and the remaining four represented people's illnesses or disabilities. Table 1 shows a sample of 10 users with (multi) labels. We focused on these labels since these were deemed to most significantly represent the social groups the medical influencers and their target audience belonged to. Where possible, we used existing taxonomies to classify users, such as Nature's and the British Encyclopedia's categorization of scientific and academic disciplines and the International Standard Classification of Occupations (ISCO) by the International Labour Organisation (ILO). It's important to note that a user can have multiple identities; for example, someone might be both a physician and a scientist. Furthermore, our categories of users did not only include individuals but also organisations.

*Social Network Construction.* We first counted and processed several types of relations, each represented as an edge in a social network graph. Users are connected by four different edges in the social network, including "Replies", "Mentions", "Retweets" and "Quotes".

Replies: Tweets are assessed for reply indicators. Upon confirmation, the "Replies_To" relationship is formed between the respondent and the original poster, utilizing the username of the latter.

Mentions: Mentions are extracted from tweet texts, which distinguish between retweets and non-retweets. For non-retweets, if the tweet is a reply and the first mention corresponds to the replied-to user, it is categorized as "Replies_To"; all other mentions are recorded as "Mention". In the case of retweets, mentions after the first are processed as separate "Mention" relations unless the retweet is of the user's own tweet, where all mentions are handled under the context of the retweeting user.

Retweets: The "Retweet" relationship is instantiated between the retweeter and the original tweet's author, distinguishing between mentions in the context of the retweet and new mentions added by the retweeter.

Quotes: The "Quote" relationship links the users quoted to the original authors, preserving the integrity of the quoted message.

We then used Python to query X's API and collect the user descriptions or bios of the actors the influencers engaged with. This resulted into a total of 88236 bios.

*Data Ethics.* Ethics approval was obtained from the University of Bristol Business School's ethics committee.

## 3 Methodology

We build a prompt-based, few-shot, multi-label classifier to assign one or more identities to medical influencers and the other actors in their social network. We then use BERTopic to identify discursive frames, i.e. the meanings that medical influencers assigned to a health issue. We use Sentence Embedding [19] to capture the semantic information from social media. Sentence embeddings from the biographies of medical influencers and other actors are used for identity classification, while embeddings from tweets are used for topic modeling.

*Sentence Embedding.* The problem of modelling natural language in a statistical way has been studied in many different domains, and is justified in Artificial Intelligence (AI) by the benefits of obtaining a representation of items, that is a vector of fixed dimensionality, in such a way that standard algebraic techniques can be used to perform inferences on the data. In other words, it allows for a convenient way to model and process data [11].

Many problems can naturally be cast in this setting, by defining the context between words. For example the standard technique known as word embedding [16] defines the relation between words as one of "co-occurrence", such that two words are related if they occur often in the vicinity of each other. Sentence Embedding is currently performed in a different way, through the use of neural networks [22], by training a model to classify pairs of sentences as semantically similar or dissimilar, with sentence embeddings produced as an intermediate representation[19].

In our case, sentence embedding is performed by a pretrained deep neural network that processes the bios and converts each biographies / tweets to a single numerical vector. These embedding vectors are trained to capture semantic similarity between vectors,

so that bio texts or tweets that have similar meanings should have similar vectors.

## 3.1 Identity Classifier

Since the majority of labels represented job occupations and organisation categories, and that organisation categories also shared similar traits with occupational categories, we trained a machine learning classifier that could accurately predict job occupations. Our overall question is: How well can we predict occupations from twitter bios by leveraging recent Natural Language Processing (NLP) approaches, such as few-shot classification and pre-trained sentence embeddings? To answer this, we used the following methods to train a few-shot classifier.

*Prompt-based Few-shot Classifier.* A few-shot nonlinear classifier is designed to require significantly fewer training examples than conventional nonlinear classifiers. This system utilizes a pretrained masked language model capable of predicting missing words in text sequences. For operation, the classifier receives a biography text followed by a structured prompt like 'The occupation is {MASK}'. It then identifies the most appropriate occupation to fill the '{MASK}' from a predefined list.

Although the model's initial training allows it to predict missing words effectively, its performance can be enhanced through fine-tuning. This process involves retraining the model with a slight modification to its weights, leveraging the minimal available data. The method is the basis for approaches such as pattern exploiting training (PET) [20].

*Multi-label Classification.* We adapt our approach to accommodate multiple labels. Our strategy involves two key steps:

(1) Firstly, during each training epoch, we substitute the mask with a randomly selected occupation from the actual set of job titles for that user provided in the training data. This promotes model familiarity with various possible occupations.
(2) Secondly, at inference time, we introduce a probability threshold, denoted as $\alpha$. Occupations with predicted probabilities exceeding this threshold are considered valid labels for the individual. This method ensures a dynamic and inclusive classification of complex identity profiles.

*Ensembles of Few-shot Classifiers.* Ensemble learning combines multiple models to improve the overall performance of a classifier. In our case, the ensemble combines different few-shot classifiers, each with a different prompt. Finally, the outputs of the individual few-shot classifiers are aggregated to make a final decision.

## 3.2 Topic Modelling

As mentioned previously, we aim to identify medical influencers' discursive frames from their tweets. We apply two different methods to discover the topic from tweets.

*LDA.* We first consider a traditional topic modelling model: Latent Dirichlet Allocation (LDA) [4]. LDA computes two different probabilities: 1) the proportion of each topic present in each document, which is a complex interaction where the presence of certain words can increase or decrease the probability of certain topics. For instance, if a document contains many words commonly associated

with a specific topic, the probability of that topic being present in the document increases; 2) The word probabilities for each topic. Simultaneously, the model computes the probability of each word occurring given a particular topic. This probability is informed by how often each word appears in documents that are associated with each topic. Essentially, if a word most frequently appears in documents where a particular topic is dominant, then that word will be an indicator of the presence of that topic.

*BERTopic.* We also explored a technique utilizing Pretrained language model, specifically BERTopic [10]. The BERTopic method can be broken down into a series of steps aimed at clustering and generating topic representations: 1) Generating sentence embeddings for each tweet to capture semantic information; 2) Implementing dimensionality reduction to transform the high-dimensional embeddings into a more manageable, lower-dimensional space conducive to clustering; 3) Clustering the reduced embeddings using the HDBSCAN algorithm; 4) Tokenizing the sentences to prepare them for analysis; 5) Determining topic representations for each cluster by amalgamating all documents within a cluster into a unified document, followed by generating a bag of words and computing the class based c-TF-IDF values for each cluster. This methodology facilitates a structured approach to identifying and representing topics within data.

We choose three different methods from BERTopic [10] to find the representative words for each cluster:

(1) Bag-of-Words with c-TF-IDF: Keywords for each topic are quickly identified, enabling easy and fast updates post-training without the need for re-training.
(2) KeyBERTInspired (Semantic Fine-Tuning): Topics are refined after the application of c-TF-IDF by analyzing the semantic relationships between keywords/keyphrases and documents. Representative document sets are created using c-TF-IDF, and topic embeddings are updated. The similarity between keywords and the topic embedding is then assessed using the same embedding model.
(3) Large Language Models for Text Generation: Topics are fine-tuned using models such as ChatGPT or GPT-4 [1]. Prompt engineering is employed, utilizing prompts including "[KEYWORDS]" and "[DOCUMENTS]" to tailor outputs based on the topic's keywords and most representative documents.

## 4 Experimental Results

## 4.1 Identity Classifier

We used a pre-trained model Albert-large-v2, from HuggingFace Transformers. This model is far more compact than widely used models like BERT-large, which uses 18x more parameters, resulting in much higher computation costs, but retains competitive performance [15]. It was shown to perform well as a basis for PET by Schick and Schütze [21].

We divided the data into training and test sets with a ratio of 80% to 20%. Each prompt was trained for 200 epochs, setting the inference probability threshold $\alpha$ to 0.1. We held out the test set until the training is complete, then evaluate the final variant of each method on the test set.

**Table 2: Comparison of different prompts for test set performance.**

| Prompt | Prec. | Recall | F1 | Acc. |
|---|---|---|---|---|
| My occupation is {} | 0.687 | 0.718 | 0.677 | 0.505 |
| This person's job is {} | 0.663 | 0.696 | 0.655 | 0.490 |
| The previous text describes a {} | 0.688 | 0.735 | 0.684 | 0.505 |
| I am a {} | 0.664 | 0.717 | 0.664 | 0.485 |
| {} | 0.661 | 0.714 | 0.659 | 0.473 |
| Occupation: {} | 0.674 | 0.726 | 0.675 | 0.495 |
| Ensemble | 0.700 | 0.752 | 0.700 | 0.513 |

We used 6 different prompts along with an ensemble method to evaluate their performance on the test set as shown in Table 2. We observed similar performance across the different prompts, with mean average precision ranging from 0.661 to 0.688, mean average recall from 0.696 to 0.735, mean average F1 score from 0.655 to 0.684, and mean accuracy from 0.473 to 0.505. The ensemble method, which combines all prompts, demonstrated superior performance, achieving the highest precision (0.700), recall (0.752), F1 score (0.700), and accuracy (0.513).

We also evaluated the robustness of the model by using different random seeds. The results show that we can achieve similar performance using different random seeds. However, we've observed that changing the random seed for splitting the train/test set may require adjustments to the learning rates. If the learning rate is improperly set—either too high or too low—the model tends to predict outcomes biased toward the most dominant classes.

The learning rates were set between 5e-4 and 1e-5, with a scheduler introduced to adjust the learning rate dynamically. During the warm-up phase, the learning rate increased linearly from 0 to the target value, and after the warm-up phase, it decreased linearly from the target value to 0 over the remaining steps. Adjustments to the learning rate (default: 1e-4) were made only if the model collapsed into predicting only majority classes. A further discussion will be continued in 5.1.

## 4.2 Topic Modelling

To better understand the semantic meaning of each tweet, we first preprocessed the data by removing the term "RT" (indicating a retweet) and deleting usernames that follow the "@" symbol (mentions). Given that the tweets are in multiple languages, we utilized the "paraphrase-multilingual-MiniLM-L12-v2 [1] sentence transformer model to generate embeddings for each tweet. We also configured the analysis settings to manage the topic granularity, setting a minimum topic size of 50, a maximum of 15 words per topic, and a unigram range.

In total, 665 topics were identified. However, 213343 tweets were categorized as noise, i.e., not associated with any topic. This large number may be because many tweets are brief and lack contextual depth, meaning they do not contain meaningful content. However, it's not certain that this is the reason for all outliers.

Many could be outliers simply because they differ significantly from the other tweets we analyzed. Among the substantive topics, the most prevalent is related to vaccine hesitancy, labelled "hesitancy_vaccins_vaccinatie_vacunas" which includes 8919 tweets. The second most significant topic, focusing on equitable vaccine distribution, "covax_equitable_vaccinequity_oneworldprotected", includes 6860 tweets. The third, focusing on mask wearing, is termed "n95_surgical_cloth_wearing" with 5638 tweets.

Figures 1a and 1b illustrate a comparison between the top topics identified by LDA and those by BERTopic. The topics derived from LDA appear more general and lack specific meaning, whereas the topics from BERTopic are notably more specific and carry clearer semantic significance. For example, the BERTopic model shows either the "Hesitancy" or the "Equity" of the vaccine (topic 0, 1), while the LDA model only demonstrates a general information in topic 0. LDA model may also identify tweets with other languages as a topic (topic 2), while the BERTopic model can capture more semantic information regardless of the languages.

Table 3 shows the three different topic representations generated from the same clusters by three different methods: Bag-of-Words with c-TF-IDF, KeyBERTInspired and ChatGPT in Section 3.2. The Keyword Lists from Bag-of-Words with c-TF-IDF and KeyBERTInspired provide quick information about the content of the topic, while the narrative Summaries from ChatGPT offer a human-readable summary but may sacrifice some specific details that the keyword lists will provide.

## 4.3 Map Relationships Between Actors' Identities, Social Network Ties, and Frames

We are interested to find additional links in the social network by leveraging the knowledge from the LLMs. Our interest extends to understanding the dynamics of message transmission within this network. The identity classifier estimates the likelihood of an individual assuming multiple identities, while the frame (topic) identification model calculates the probability that each word embedding corresponds to a specific frame. These probabilities are averaged over each tweet to associate tweets with various frames.

Utilizing the outputs from these models, we aim to construct a new graph where nodes represent not just Twitter users, but also the frames (topics) they discuss. Each user node will also feature an attribute detailing their identities, which defines the influence of medical professionals within their network and how their messages resonate across various user communities.

Figure 2 illustrates the exchange of messages within this social network. For instance, the scientist "MedInf3" engages in discussions about the emergence of a virus with another scientist, "Actor7". Simultaneously, "MedInf3" also converses on the same topic with a cultural professional and scholar, indicating the cross-disciplinary nature of the discussion. Additionally, "MedInf3" broaches a broader topic, "pandemic", with "Actor5", whose identity is not predictable from their biography. This visualization aids in exploring how the perspectives of medical influencers on health issues proliferate across social media communities.

Figure 3 shows a straightforward message passing process within a socio-semantic network, illustrating how the message circulates between MedInf1 and MedInf3.

---

[1]model available at: https://huggingface.co/sentence-transformers/paraphrase-multilingual-MiniLM-L12-v2

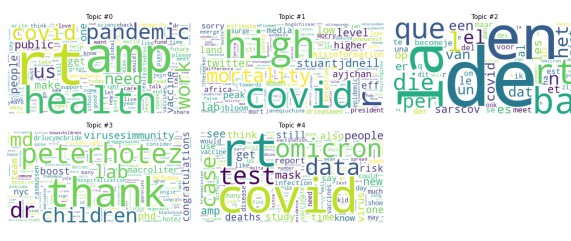

(a) LDA top 5th topics

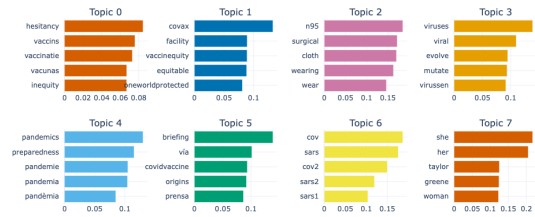

(b) BERTopic top 8th topics

**Figure 1: Comparisons of LDA topics and BERTopic topics. The topics from BERTopic are more semantically coherent.**

**Table 3: Comparison of Three Different Topic Representations Methods of BERTopic with the Same Clusters**

| Topic | Count | Original Representation | KeyBERTInspired | ChatGPT |
|---|---|---|---|---|
| 0 | 8919 | [hesitancy, vaccins, vaccinatie, vacunas, inequity, patents, vaccineren, patent, vaccinate, rollout, distribution, hesitant, vaccin, vaccinating, equity] | [vaccineren, vacunar, vacunats, vacunarse, vacunas, vacunación, vacuna, vaccins, vaccin, vaccineequity] | Importance of Vaccine Equity and Distribution |
| 1 | 6860 | [covax, facility, vaccinequity, equitable, oneworldprotected, african, rollout, delivered, vacunas, ghana, covidvaccines, covidvaccine, donated, ensure, income] | [covid19vaccines, covid19vaccine, vaccinequity, vacuna, vacunación, vaccinating, vacunas, vaccinate, vaccins, vaccinations] | COVID-19 Vaccine Rollout and Equity |
| 2 | 5638 | [n95, surgical, cloth, wearing, wear, kn95, ffp2, n95s, mask, filtration, fit, masks, masking, worn, bettermasks] | [maskers, masks, masken, maske, mask, masking, masked, masques, wearers, masque] | Importance of N95 and Eye Protection in COVID-19 Prevention |
| 3 | 5624 | [viruses, viral, evolve, mutate, virussen, evolution, rna, load, host, mutations, loads, norovirus, humans, replication, replicate] | [virussen, viruswaanzin, viruses, virusfeiten, viral, norovirus, arboviruses, antiviral, antivirals, arbovirus] | Evolution and Mutation of Viruses in Human Hosts |
| 4 | 4936 | [pandemics, preparedness, pandemie, pandemia, pandèmia, throughout, pandémie, prepare, lessons, ending, future, preparing, inevitable, pandem, failures] | [pandemiewet, pandemictreaty, pandemias, pandemien, pandemico, pandemi, pandemics, pandemie, pandemia, pandem] | Public Health Crisis and Pandemic Preparedness |

## 5 Discussion

This study presents methods to identify and analyze the identities of actors, illustrating how medical influencers adapt their discourse to reach and influence different audiences.

### 5.1 Identity Classifier

1. What is the best prompt? We can witness similar performances across different prompts after enough epochs of training, however, the prompt including the word "occupation" performs slightly better than the more naturalistic phrases. Overall, the result for the Ensembles of 6 different prompts is substantially stronger performance, as the ensemble benefits from the "wisdom of the crowd" of the diverse set of classifiers: each classifier makes different mistakes, so that typically, the majority of classifiers are correct. This is important particularly for a multi-label classification task since users' identities might not only relate to one social group (e.g. occupation) but also to other social groups (e.g. being a parent or have a illness or disability).

2. Why might different settings of random seeds for train/test splits require different learning rates? One possible reason is the imbalances in the dataset, where some classes are over-represented compared to others, can require adjustments to the learning rates. For instance, a dataset that ends up with a disproportionate number of samples from dominant classes can lead the learning process to converge too quickly, in this case, a smaller learning rate will be needed. Conversely, a more balanced or differently distributed test set could expose the model to a wider variety of data scenarios, and in this case, a larger learning rate is needed.

### 5.2 Topic Modelling

1. Do pretrained models help performance of topic modelling? When comparing embedding-based topic clustering methods like BERTopic with traditional Latent Dirichlet Allocation (LDA), BERTopic demonstrates significant advantages due to its use of pretrained models. These advantages include a deeper understanding of semantic context, which is especially beneficial for accurately clustering topics in shorter texts such as social media content. BERTopic also excels in providing more interpretable topics and offers greater flexibility with adjustable parameters to tailor the model to specific needs. Additionally, its capability to handle multilingual data makes it a versatile choice for global applications, setting it apart from the more rigid, length-dependent LDA approach.

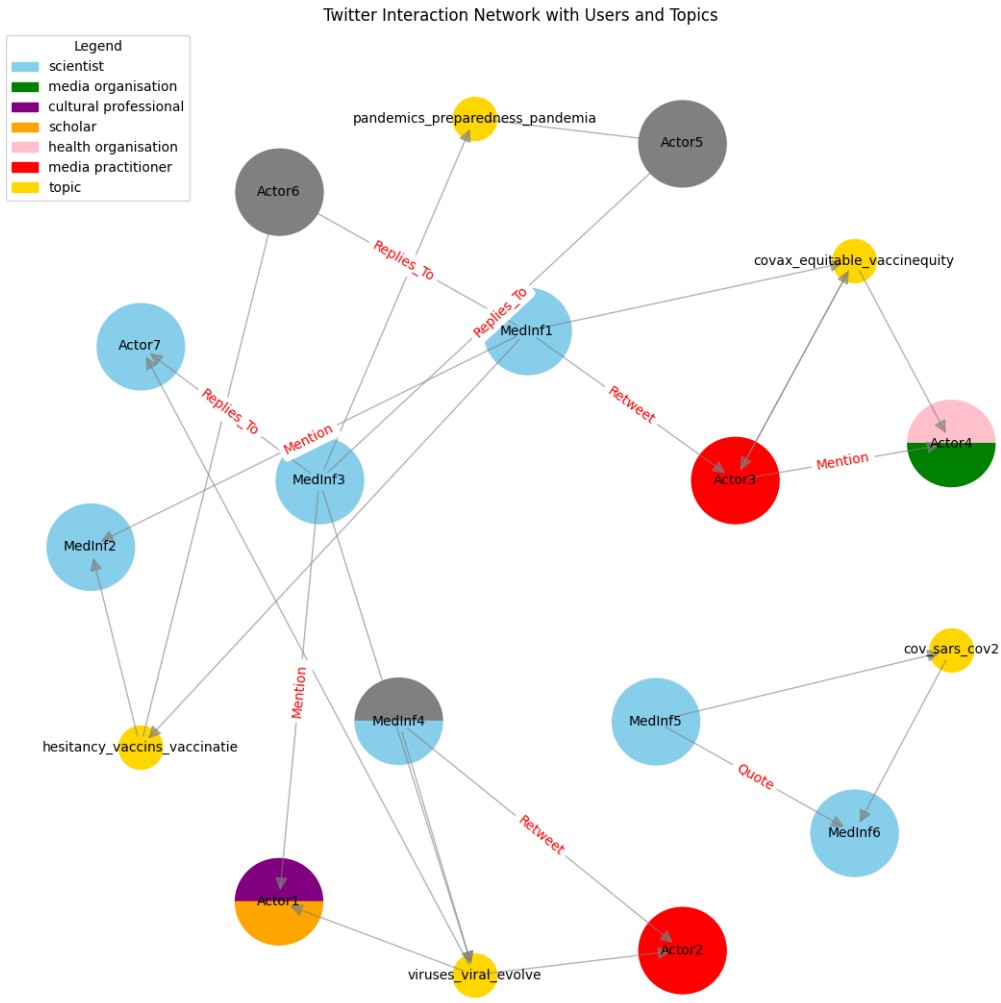

**Figure 2: Message Exchange Among Social Network Actors**

2. Given that the clustering method HDBSCAN is an unsupervised method, how to control the number of topics? Comparing with Automatic Topic Reduction during training, if we initially retain the original clusters formed by HDBSCAN, the topic clusters are more meaningful. This enables us to first assess the breadth of topics generated and then make informed decisions on their refinement. Following this assessment, we can manually refine or reduce the number of topics based on insights from the hierarchical clustering output, tailoring the final topic set more precisely to your analytical needs.

3. How to find the representative words of the topic? One of the key elements of BERTopic is its use of a Bag-of-Words model and c-TF-IDF for weighting (Figure 3). Our tests on different n-gram lengths show that unigrams yield the most accurate topic

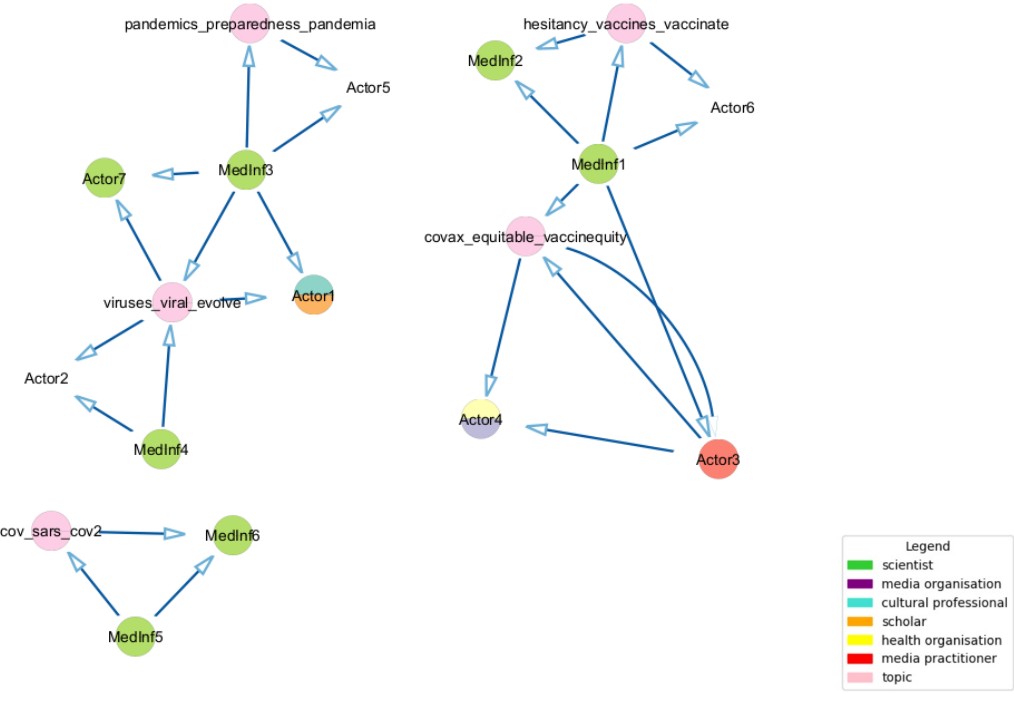

**Figure 3: An illustration of message passing, showing the connections between users of diverse identities through various topics.**

representations. However, we have observed that the descriptions generated by the ChatGPT model depend heavily on the chosen representative documents, which compromises their accuracy.

### 5.3 Future Work

Our future work will focus on enhancing our identity classification by experimenting with various prompt configurations and refining our topic models, for instance, through manual adjustments using hierarchical clustering. To answer this, we investigate some more detailed questions:

1. Do pre-trained models help performance, given that there is a large difference between the language used in Twitter bios and the text data used to pretrain the models? It is possible that simpler models that learn associations between n-grams and occupations may perform better than few-shot and sentence embedding methods given the domain shift. Hypothesis: pretrained models outperform n-gram models, since there are many words and n-grams that will not be seen in training for every class.

2. Can ensembles of classifiers help since this has been shown in previous work to be effective for combining different few-shot classifiers? Hypothesis: there will be a small gain from combining

classifiers that use different prompts, since it is difficult to design a single prompt that works effectively on all examples, but the diversity of models will be quite limited.

3. Are there some occupations that we can predict more reliably, and some that are difficult to detect? We may find that some occupations are rarely recalled, and that certain occupations are frequently chosen incorrectly. We may be able to identify possible reasons for failing to detect certain occupations by inspecting some examples for each error type. Hypothesis: strong variation in performance of the classifier between occupations.

4. Additionally, we plan to conduct a network analysis to explore the relationships and interactions between identified entities. This analysis will involve mapping connections within the data, examining the influence and reach of specific nodes, and potentially discovering hidden patterns or communities within the network.

## 6 Conclusion

In this study, we have demonstrated techniques to map the identities and topics of actors, showing how medical influencers tailor their discourse to impact diverse audiences. Specifically, we utilized

a multi-label few-shot classifier based on prompt learning to categorize user identities. We then employed BERTopic to identify the common topics or frames shared among these influencers within the network. Subsequently, we mapped the identified identities, social network connections, and discursive frames onto a socio-semantic network. This innovative approach in socio-semantic network analysis allows us to understand how the identities of medical influencers influence their positioning within the network and the extent to which their messages resonate within user communities.

## Acknowledgments

This project has been supported by the Jean Golding Institute for data science and data-intensive research at the University of Bristol. We would also like to thank Martha Lewis for the travel fund, provided through her Faculty of Engineering Pump-Priming funds from the University of Bristol.

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
