# OpenReview forum: "Medfluencer: A Network Representation of Medical Influencers’ Identities and Discourse on Social Media"
_KDD.org/2024/Workshop/epiDAMIK — KDD 2024 Workshop epiDAMIK_

### Official Review · Reviewer_SbaC · 2024-06-28
**Great insight and interesting work with potential for improvements**

**Rating:** 4
**Confidence:** 5

**Review:**

The authors study the problem of inspecting how medical influencers can impact the social graph during crucial events such as pandemics. While this is a well studied and actively researched topic, the authors bank on the hypothesis that the "identity" of a medical influencer is an important attribute in constructing such social graphs.

Detecting "identity" of social users in terms of their group identity, occupation identity, and personality traits is a non-trivial problem as these are often expressed from users actions in the network. In this paper, the authors proposed a multi-label classification model to detect identities from user bios. The authors conducted human labeling of a small set of users and show promising performance on the set to justify the module. Crucially, the authors acknowledge some of the limitations with small data size in this experiment.

The authors further use a pre-trained LM to identify topics (BERTopic) on user interactions to identify how and what topics the influencers converse on and how these are spread through the social network. In addition to centrality of the graphs, the authors in the paper show qualitative examples of how user identities can play a role in the spread of specific discourse

Overall, the authors identify some interesting patterns and given the page limits does an admirable job of conveying some of the key results.

However, for a full archival format, the paper can be further improved upon by addressing some of the following areas:
-  The authors primarily considered the social graph and influencer bios to be a static one. However given the period of study (the covid-19 pandemic), the interactions may be strongly influenced by external events that may have been captured in temporal deviations from the average. It would be useful for the authors to either discuss this limitation or propose temporal variations of the method
- On a different angle, while some case studies have been presented the importance of the enriching the social graph with user identity seemed limited. For example, it would be provide more details on how specific patterns such as vaccine hesitancy spread based on influencer identity and whether any actor correlation was observed

---

### Official Review · Reviewer_Suqs · 2024-06-29
**Well-written paper on building a network representation of the role of medical influencers using identities, topics and interactions.**

**Rating:** 5
**Confidence:** 3

**Review:**

This paper deals with building a socio-semantic network to analyze the role of medical influencers in shaping the discourse on social media. Their effort is to map actors' identities to the topics they discuss and their social media interactions. The application area (i.e. medical influencers' role) is justified in the paper as being of public health significance.
- The paper is clearly written and the methodology is described in detail.
- The novelty stems from considering the actors' identities and their discursive frames in the network representation. They use prompt-based few-shot learning for the identities of the actors from their bios. The work's novelty could be better ascertained with a related work section.

Pros:
1. Significance: The work makes progress towards better understanding the influence wielded by medical personalities on social media.
2. Clarity
3. Novelty: the work considers the identities of the actors and ties these to health topics they discuss.

Suggestions:
1. A figure of the framework would help to show where each component goes in their overall method at a glance.
2. The paper could include some discussion on the insights from their constructed networks such as from community analysis, centrality measures, exploration of how identities relate to influence etc. (as mentioned in their future work)
3. Related work section would be a good addition.
4. Some argument for advantages of socio-semantic network over social network in assessing influence. What are the trade-offs?

---

### Official Review · Reviewer_ckSn · 2024-06-30
**Interesting paper with a detailed methodology**

**Rating:** 4
**Confidence:** 4

**Review:**

The authors propose a few-shot multi-label classifier to categorize medical influences on social media and a pipeline to create a map highlighting how these actors interact with each other and with different COVID-related topics.

Pros:
•	The methodology is clear and detailed.
•	Various topic representations have been evaluated and presented clearly.

Major comments:
•	Future works mention mostly the performance of the predictive model. Instead, developing the applications of the map to prevent misinformation would be preferable.

Minor comments:
•	Please provide a GitHub repository of the codes to allow replication of the analysis.

---

### Official Review · Reviewer_LiFo · 2024-06-30
**Promising ideas, but needs stronger performance and deeper results**

**Rating:** 2
**Confidence:** 4

**Review:**

This paper studies the tweets and social network of the top 100 medical influencers on X (formerly Twitter) during the COVID-19 pandemic. Their goal is to conduct a socio-semantic network analysis, to simultaneously study the individuals' identities, their relations to each other, and the content in their communications. To do this, the authors take the following steps:
1) Data collection: they identify the most influential healthcare professionals on X and collect tweets made from these accounts from Dec 1, 2020 to Feb 24, 2022
2) They construct the social network between the influencers and the actors they engaged with, with different types of edges (Replies, Mentions, Retweets, Quotes)
3) They build an identity classifier, using a pretrained language model, to classify the job occupation of each user based on their bio (allowing users to have multiple occupations)
4) They apply topic modeling techniques, such as LDA and BERTopic, to learn topics from the tweets

Strengths
- The proposed analysis is interesting: it would be useful to understand what medical influencers are talking about during the pandemic and the structure of the network
- Several steps are completed, including data collection, identity classification, and topic modeling

Weaknesses
- The performance of the identity classifier is not very strong (best model achieves acc of 0.64) and very inconsistent across metrics (F1 vs acc). The F1 systematically improves from cross-val to test (sometimes over 20 points) but the accuracy systematically worsens from cross-val to test (also sometimes over 20 points). The authors comment on this in their discussion, but their reasoning then calls into question how much we can trust the reported numbers (they primarily point to possible distribution shift from cross-val to test, but why does this distribution shift happen, why are they different distributions if they are drawn from the same dataset, etc)
- I would like to see far more insights derived from the analysis. The primary results seem to be the performance of the identity classifier and the top 5-8 topics from the topic models, presented as top words per topic. Several prior works have already done topic modeling over COVID tweets (eg, https://www.jmir.org/2020/11/e20550/, https://www.jmir.org/2020/10/e22624/, https://www.jmir.org/2021/6/e24435/), so it would be useful to deepen this analysis. For example, how do the topics relate to the structure of the network? How do topics in this dataset differ, given that it focuses on influencers, compared to topics from general populations? What is the structure of the reply trees for different topics? Is there evidence of diffusion? I think there's an opportunity for interesting analyses here, given the setup of identities+network+topics, so I encourage the authors to dig deeper.